# Qualitative study exploring which research outcomes best reflect women's experiences of heavy menstrual bleeding: stakeholder involvement in development of a core outcome set

Natalie Ann MacKinnon Cooper ![ORCID],[1] Sarah Yorke,[1] Alex Tan,[1] Khalid Saeed Khan ![ORCID],[2] Carol Rivas ![ORCID] [3]

¹Women's Health Research Unit, Wolfson Institute of Population Health, Queen Mary University London, London, UK
²Public Health, Faculty of Medicine, University of Granada, Granada, Spain
³Social Research Institute, UCL Institute of Education, London, UK

**Correspondence to**
Dr Natalie Ann MacKinnon Cooper;
natalie.cooper@qmul.ac.uk

## ABSTRACT

**Objective** This work contributed to the development of a core outcome set (COS) for heavy menstrual bleeding (HMB). The objective was to determine which research outcomes best reflect how HMB affects women's lives and to identify additional research outcomes, not previously reported. It was important to explore and record participants' reasoning for prioritising outcomes and use this information to reinforce the patients' voice during later phases of the COS development.

**Design** Patient workshop discussions and telephone interviews.

**Setting** East London teaching hospital.

**Participants** Inclusion criteria were that participants must be over 18 years old, that either they or their partner had a history of HMB and that they had a good understanding of written and spoken English.

**Results** 41 participants were recruited for the study. 8 women and 1 man completed the study. The eight female participants were representative of the different underlying causes and treatments for HMB. Participants ranged in age from their early 20s to their 60s and represented a range of ethnic groups. The five main themes that were identified as being important to patients were: *'restriction', 'relationships and isolation', 'emotions and self-perception', 'pain'* and *'perceptions of treatment'*. We identified eight coding nodes that did not correspond with our list of previously reported outcomes in studies of HMB. These nodes were consolidated and became five new outcomes for potential inclusion in the COS.

**Conclusions** HMB stops women living their lives as they would wish. It affects their relationships, education, careers, reproductive wishes, social life and mental health. This is a condition of girls and women in the prime of their lives, but for many, the constant threat of a heavy period starting means that they sacrifice that freedom. The societal and economic costs of women being incapacitated every month has an effect on everyone.

**Trial registration** The COS study is registered with the COMET (Core Outcome Measures in Effectiveness Trials) Initiative—project reference number 789.

## STRENGTHS AND LIMITATIONS OF THIS STUDY

⇒ All the common underlying causes of heavy menstrual bleeding (HMB) were represented by women in our workshop and interviews and the group also reflected a range of ages and ethnicities.
⇒ We were limited when exploring the partner's opinions by only having one partner attend our workshop, sadly reflecting the common view that periods are 'women's business'
⇒ Data saturation was reached and was in keeping with international workshop data, confirming the validity of our results, despite the high attrition rate.
⇒ Women felt heard and expressed their satisfaction about talking openly about HMB.

## INTRODUCTION

Heavy menstrual bleeding (HMB) affects approximately one in four women[1]; however, despite being prevalent it is not a prominent topic in society and many women suffer in silence.[1] HMB is a symptom with multiple underlying causes[2] and numerous types of intervention, for example, hormonal, medical, surgical,[3] thus it is important that trials of interventions for HMB produce data that allows treatment options to be easily compared. In general, women have been hard to recruit to research studies,[4–6] and only a small amount of annual UK research funding (approximately 2.5%) is spent on reproductive and childbirth conditions.[7] It has been suggested that up to 85% of health research goes to waste[8] because it is not reported, reported badly or does not adequately build on existing research. So, when research into HMB does get funded, we need to ensure it asks the right questions and is conducted rigorously.

COS are a disease-specific agreed set of outcomes that are established as a reporting

standard minimum for all relevant clinical trials. The aim of COS is to ensure that studies of a condition all report the same, valid outcomes which will ultimately mean they will produce results that are not only useful for interpretation of that trial but also allow easy data synthesis and comparison. The Core Outcomes Measures in Effectiveness Trials (COMET) initiative[9] developed standardised methodology for development of COS. A search of the COMET database revealed that no COS existed for HMB. The only previous work exploring outcome reporting in trials of HMB was restricted to primary outcomes in randomised controlled trials.[10] This study identified that there was a lack of consistency in reporting endpoints and tools and that outcome standardisation was required. With that in mind, we set about developing a core outcome set (COS) for HMB to improve the quality of future research.

COMET methodology has four phases: (1) a literature review of previously reported outcomes; (2) qualitative work with patients to identify which outcomes are most important to them and identify any outcomes not previously reported; (3) a consensus survey involving all key stakeholders; (4) a consensus meeting to finalise the final COS. This paper describes the qualitative work which we undertook in phase two of the HMB COS development. The aim was to identify how HMB affects women's lives, how this relates to previously reported research outcomes and to identify previously unreported outcomes. We identified the importance of generating transcripts from this work to inform discussions during later stages of the COS development to ensure that patients' views remained heard.

## METHODOLOGY

Phenomenological methodology underpinned the study with the aim being to understand the most important effects of HMB on the lives of women—their phenomenological experiences—to identify what research outcomes might be most important to them.

### Recruitment

This study recruited a purposive volunteer sample of women with HMB and their partners to be representative of different ages, ethnic backgrounds, underlying HMB cause and treatments received.

Inclusion criteria were that the potential participant should be aged over 18 years, that they or their partner had a medical history of HMB and that they had a good understanding of written and spoken English. This latter criterion was necessary because our resources meant we could only hold one workshop and therefore all participants had to be able to discuss the topic together. Participants were recruited from gynaecology clinics in a teaching hospital. Clinicians identified people, including partners, who fulfilled the inclusion criteria. The definition of HMB is subjective as per the UK National Institute for Health and Care Excellence (NICE) guideline, the International Federation of Gynecology and Obstetrics

(FIGO) and the American College of Obstetrics and Gynecology (ACOG).[3 11 12] Thus, having been diagnosed in the National Health Service (NHS) clinic was enough to be included and was based on the subjective definition in the NICE guideline. A member of the research team went through the full informed consent process with each person who expressed an interest. The contact details of participants were stored on a secure database. The maximum number of participants considered feasible for a workshop with breakout discussions was 30,[13 14] so we aimed to recruit up to 40 participants to allow for dropouts.

We used a workshop rather than a focus group format to discuss divergences and similarities among a heterogenous group and develop group consensus on the outcomes to take forward.[14] Focus groups are not intended to reach consensus, but to reveal group norms within a homogenous group, a different concept.[15]

Once recruitment was completed, we emailed the participants to identify what day and time would suit most participants and scheduled the meeting accordingly. The meeting was held at the same site where women were recruited and so was a feasible place for them to travel to and at the same time provided a comfortable, non-threatening space. Prior to the workshop, participants were contacted by telephone to confirm their attendance.

### Workshop

The workshop was conducted by three female researchers (authors NAMC, CR and Johana Nayoan (JN) a research fellow in health behaviour). CR and JN were both experienced in mixed methods women's health research. The professional background of the researchers and the reasons for the study were introduced before the start of the session. One researcher introduced the workshop, a second led the whole group discussion and all three were involved in the small group discussions/interview. In addition, a male research fellow (AT) and a female research assistant (SY) were present to help with set-up, administration and refreshments. Prior to the workshop, the participants would have met one of the researchers (NAMC, SY or AT) during recruitment. The workshop was held in spring 2018 so there was no requirement for social distancing measures.

The workshop started with an introduction by NAMC who led the COS HMB project. She re-explained the concepts from the patient information sheet about 'outcomes' in research studies and the problems caused by variation in outcomes. She then explained the workshop format and reminded participants the discussions would be audio-recorded. The workshop was designed to split up participants for small group discussion, based on the underlying cause of their/their partner's HMB, for homogeneity of symptoms and treatments. A facilitator was assigned to each group and used a list of prespecified questions (see online supplemental appendix 1) to guide discussion about the effects of HMB and its treatment on their lives. The small groups were also asked to try and

prioritise which outcomes would be most important for them if they were being offered a treatment. The groups then came together as a whole to feed back to each other their main issues and prioritised outcomes. These were listed on a flipchart and then one researcher led discussion among the whole group about each suggested outcome to explore the full range of participants' opinions and gain a true consensus. The workshop lasted approximately 2 hours, with a refreshment break between the small group and whole group discussions.

Participants were reimbursed for their travel and following the workshop received a £20 voucher to thank them for their participation.

## Telephone interviews

We emailed the recruited participants who had not attended the workshop, asking if they would be willing to undergo a telephone interview. Those who agreed were interviewed by the clinical academic researcher (NAMC) as she had recruited each of these women to the project; thus they had met prior to the interview. The interviewer used the same list of prespecified questions that was used at the workshop to guide discussion. Interviews lasted between 10 and 30 min. The interviews were audio-recorded with the women's consent. These participants also received a £20 voucher to thank them for their time.

## Analysis

The workshop and interview recordings were transcribed and the transcripts checked against the audio recordings for accuracy; participants were not asked to review and correct them. Identifying information was removed from the transcripts and the original recordings were deleted.

NVivo V.11 software was used for data management. Thematic analysis was conducted. Two researchers familiarised themselves with the data and coded them independently for themes. Discrepancies were then discussed and recoding continued until consensus (≥75% agreement) was achieved.

To identifying additional outcomes for development of our COS, we examined the list of codes and excluded those that could not be used as research outcomes because they referred to aspects that would not be modified by treatment (eg, age, race, outcome priority). We then cross-referenced the remainder against the list of outcomes from our systematic review of the literature to identify any not previously reported. These codes were then reviewed and combined where relevant, to determine new outcomes to be added to our list for the next step of the COS development, a consensus survey.

We further explored the reasoning behind the themes to try and identify why participants had emphasised the issues raised.

## Patient and public involvement

A patient and public advisory group (Katie's Team) were involved in developing the consent form, patient information leaflet and data information sheet for recruitment to the study. Patients were the main participants in this work.

## RESULTS

### Participation

A total of 41 people were recruited to the project. A total of 16 participants confirmed that they would attend the workshop; however, only 6 attended on the day. Of these six, one participant was a male partner of a woman with HMB and the remaining five were all women who had HMB or had been treated for it. Two women had fibroids as the underlying cause of their HMB, one had endometrial hyperplasia and two women reported an endometrial or ovulatory cause (one of these women had also had an endometrial polyp which can contribute towards HMB). Because of the small number of attendees, our breakout groups were smaller than is usual. However, to ensure that no one participant dominated the discussion and skewed the data towards their underlying pathology, we decided to still hold them prior to the whole group discussion. The male participant was interviewed separately in parallel, by one of the research team, using the same questions, as he was likely to have a different perspective from the patients.

Three participants who could not attend the workshop agreed to undergo telephone interviews; one had endometrial hyperplasia, two who had an endometrial or ovulatory cause for their HMB. No new themes emerged from the interviews, so we were satisfied that data saturation appeared to have been reached.

Table 1 shows details of the workshop and interview participants.

The eight female participants were representative of the different causes and treatments for HMB and as they ranged in age from in their early twenties to in their sixties, they also represented women at different points in their reproductive lives. Participants represented a range of ethnic groups.

### Coding and themes

Data were coded using 88 different concrete nodes that were determined inductively. The most commonly coded nodes were 'pain' (coded 77 times and identified in 7/7 transcripts), 'relationships' (coded 50 times and identified in 7/7 transcripts) and 'heavy bleeding' (coded 46 times and identified in 7/7 transcripts). We used the nodes to identify new outcomes for development of our COS and we supplemented this by identifying themes within the transcripts to give us a deeper understanding of why certain aspects were so important.

### Nodes as outcomes

A total of 20 nodes were excluded as they could not be used as outcomes or assessed as outcomes. A total of 60 nodes were considered to already be present in our list of outcomes. A total of 8 nodes were not identified on

**Table 1** Data regarding the workshop and interview participants age, ethnicity and the underlying cause of their heavy menstrual bleeding

| Participant | Age | Ethnic group* | Cause of HMB (PALM–COEIN classification[24]) | Types of treatment received |
|---|---|---|---|---|
| Workshop participant 1 (female) | 20–30 years | White British | Endometrial or ovulatory (AUB-E/AUB-O) | Hormonal and hormonal coil |
| Workshop participant 2 (female) | 60–70 years | White British | Endometrial hyperplasia (AUB-M) | Hormonal coil |
| Workshop participant 3 (female) | 30–40 years | Asian British | Endometrial or ovulatory (AUB-E/AUB-O) and endometrial polyp (AUB-P) | Hormonal, hormonal coil and surgical |
| Workshop participant 4 (female) | 50–60 years | Caribbean | Fibroids (AUB-L) | Surgical |
| Workshop participant 5 (female) | 40–50 years | African | Fibroids (AUB-L) | Surgical |
| Workshop participant 6 (male) | 20–30 years | White British | Partner of participant workshop 1 | N/A |
| Telephone interviewee 1 (female) | 40–50 years | Black British | Endometrial hyperplasia (AUB-M) | Hormonal coil |
| Telephone interviewee 2 (female) | 30–40 years | Caribbean | Endometrial or ovulatory (AUB-E/AUB-O) | Hormonal |
| Telephone interviewee 3 (female) | 30–40 years | Asian British | Endometrial or ovulatory (AUB-E/AUB-O) | Hormonal |

*As per UK government 2001 census definitions, https://www.ethnicity-facts-figures.service.gov.uk/style-guide/ethnic-groups.
AUB, abnormal uterine bleeding; HMB, heavy menstrual bleeding.

our list, these were 'cost to patient', 'partner inclusion', 'relationships', 'self-consciousness', 'self-image', 'stigma', 'shame and embarrassment' and 'understanding and empathy'. One node, 'lack of understanding and support,' appeared on our list in the context of professionals but not in the context of patients' partners, so this was also included as a new outcome. By merging similar outcomes through research group discussion, we identified five new outcomes to add to our long list. Online supplemental appendix 2 shows which nodes were excluded and how the remaining nodes were matched to pre-existing outcomes. Table 2 details the newly identified outcomes and the grouping of nodes used to determine the outcome.

### Themes

From our basic nodes, we identified more conceptual themes within the transcripts. Nodes were not assigned to a theme if they pertained to specific physical symptoms which we expected to see frequently within the transcripts, for example, 'heavy bleeding'. The grouping is shown in table 3.

### Restrictions

Participants frequently talked about restrictions placed on their lives due to periods. So many aspects of life were affected: what women wear, how they perform at work, their freedom to go out, not being able to make plans, what they eat, how they sleep, worries about family and ability to exercise. There was not an aspect of life that was not affected, women even discussed how HMB put restrictions on their plans to have children. Relationships were affected in several ways and discussed so much that the effect of heavy periods on relationships is presented as a separate theme. Along with the heaviness of the bleeding, pain and irregularity were other reasons given for feeling unable to do things. Table 4 shows the variety of restrictions experienced.

**Table 2** New research outcomes and the nodes used to identify and define them

| Newly identified outcome | Nodes used to determine this outcome |
|---|---|
| Cost to patient | ▶ Cost to patient |
| Degree of self-confidence | ▶ Self-image<br>▶ Self-consciousness |
| Partner assessment of effect of heavy periods | ▶ Partner inclusion<br>▶ Relationships<br>▶ Understanding and empathy |
| Quality of relationship with partner | ▶ Relationships<br>▶ Understanding and empathy<br>▶ Lack of understanding and support (from partner) |
| Degree of embarrassment/shame | ▶ Stigma<br>▶ Shame and embarrassment |

**Table 3** How nodes linked to the emergent themes from the workshop and interviews

| Nodes | | Theme |
|---|---|---|
| Mental challenge (ie, cognitive functioning) Feeling weak Helplessness Sleep disturbance Employment and education Restriction Practicalities or logistics Life affected on no—menstruating days Responsibility, for example, for family or at work | Cost to patient Home life Childcare Irregularity Religion Freedom Changing sanitary protection Lifestyle | Restriction |
| Relationships Lack of understanding and support Sex Partner inclusion Loneliness Emotional support Isolation | | Relationships and isolation |
| Lack of understanding and support Self-consciousness Shame and embarrassment Leaking Self-image Emotions Feeling different Femininity Hopelessness Sense of control | Stigma Frustration Anxiety Mood swings Psychological effects Mental health Scared Fear Self-reflection Acceptance—of condition and pain | Emotions and self-perception |
| Pain management Pain Suffering Dyspareunia | | Pain |
| Government unsupportive GP inconsistency Accountability of healthcare staff Seeking medical help Delayed referral Adverse effects in general Treatment effectiveness Treatment preference Payoff between treatment and side effects Hormonal treatments Available resources | Treatment stopped Treatment safety Adverse effects on fertility Procedure complexity Desire for fertility Age—individualised care Race—individualised care Communication Knowledge Understanding and empathy Discrimination | Perceptions of treatment |

GP, General practitioner.

A restriction that may not be generally recognised is the financial cost to patients of having heavy periods. There has been much in the media about the taxation of sanitary products and thankfully, in the UK we saw an end to this at the start of 2021[16] but actually, sanitary products are only a small part of what women can end up paying, as explained by one woman:

> Because it's not just pads, it's clothing, sheets, towels and painkillers. (WP3)

Our group recognised that many women would not be financially comfortable enough to be able to foot these costs or to compound them by taking time off work. As a result, they were restricted from being able to manage their bleeding and were often forced to put themselves in to situations where they risked embarrassment and shame.

> It's not a pleasant thought to think that another woman is suffering the same thing but she hasn't got the same comforts or can't just buy the heating pads or can't just go home, take the day off work because she can't afford these things. (WP1)

Shame and embarrassment are emphasised within the next theme 'emotions and self-perception'.

### Emotions and self-perception

Our workshops and interviews highlighted the impact on how women felt about themselves but also the worry of what other people might think about them. If the pain and bleeding can be managed, either with analgesics or adequate sanitary protection, then women can to some extent function. It is when these issues cannot be managed, that they cannot be hidden. A common worry is blood leaking through sanitary protection and clothing, so that other people can see. This is often termed 'flooding' by UK gynaecologists. Women gave examples of when their bleeding had caused them embarrassment.

> How embarrassing it is if you have an accident and you're out. Just generally, how often you have to change. At work, it's embarrassing because I always have to carry something with me just in case. (TI1)

> And sometimes I'm scared to air my mattress outside, because somebody would think I've weed… but it's not wee, it's blood stains. (WP4)

As with 'restrictions', any aspect of irregularity or unpredictability added to fears of embarrassment as women might be taken by surprise.

> I …. got up, and there was the biggest red stain and I was covered in blood…… I was literally like scrubbing this sofa, and it wasn't mine, and other people use the sofa. And I was like in tears, because…. I didn't know what to do in case anyone came home and saw it. And I felt quite a lot of embarrassment and shame about that. I mean it was like I've actually damaged this property, and I didn't do it on purpose and that was so shameful. (WP1)

Even when the bleeding can be hidden, other symptoms associated with menstruation have an impact on how women feel about themselves.

**Table 4** Restrictions experienced by women with heavy menstrual bleeding

| Restriction | Example |
|---|---|
| Professional/academic | I was a teacher and when you're a teacher you can't really leave the classroom so if I got a heavy period, I used to have to layer myself up with so much stuff, because I knew that there was absolutely no chance of leaving… I always had to wear black trousers just in case I bled through… (WP2) |
| | I worked in retail…. and every time I had to go away from the counter… I'd be like to my female friend, 'just check me, check me'. That was constant. (WP1) |
| | For the first three to four days I can't go to work, I can't go to uni…, I can't really leave the house that much because I need to change every 25 minutes to half an hour. It doesn't really make me feel comfortable because I leak. (TI3) |
| Clothing | (thick pads) take up a lot of space in your bag, but you have to carry it or how are you going to survive? You don't want blood on your clothes and all of this. But I wear double pads. You have to put on black pants and black panties and that sort of thing,… because that kind of blood will go through panties and also a dress and all sorts. You have to very well pad up yourself… because the blood goes all the way past your bottom, and coming up. (WP4) |
| Relationships | I didn't marry him because I was afraid my fibroids (would pass on to) … the kids we made; I don't want it to go to my kids. (WP5) |
| Social | I have to book everything around my period, and it's the centre of my life. If someone's having a party, the first thing I do is ask 'what is the date?' to see if I can go or not. (TI1) |
| | I can't go to the gym again because I'm paranoid (of leaking) because I have to use the equipment while I'm sitting down a lot. It's stopped me from going to church, … I have to call off work if I need to rest. (TI2) |
| | Obviously to stop, the irregularity………., when it (bleeding) did stop she was always on edge it was going to start again. Because it was so heavy, obviously, you need to be prepared for it to start at any point which is obviously very tiring for her… it's one thing to have a painful period, but to be irregular or constant, there was no break for her at any point. (WP6) |
| Activities of daily living | When it was getting heavier and heavier and I was feeling more weakness, I was just in bed. I was not able to do anything, no housework. I was off work. So that situation was like I had some serious life-threatening condition, which is preventing me to do everyday chores. (WP3) |
| | It affects my eating because while I'm on for the whole 6 days I don't eat… literally everything I eat I just vomit it out… It affects my sleeping because I don't really sleep when I'm on unless I'm coming off and the pain is gone. Last night I was in and out of my sleep vomiting, vomiting, vomiting and I was waking up my parents. (TI3) |
| | I have to choose a toilet I know is going to flush. It is so ridiculous. I know the toilets at work that don't work very well. I know the ones that flush properly, so I know where to go. (TI3) |

It's just so uncomfortable for weeks and it's not just that you feel like that, internally it's externally. You don't feel beautiful that week. (TI2)

When we asked a woman's partner (WP6) what he thought the biggest impact was on her, he responded, "it's everything. Her image, self-worth, her confidence." Within the group, it was recognised that often a partner or loved one might make a better assessment of the severity of women's symptoms as they are able to evaluate the woman as a whole and do not feel obliged to pretend that 'it's not that bad'.

### Relationships/loneliness
Relationships are an important aspect of everyday life. Not only relationships with partners but with other family members, friends and colleagues. The impact of HMB on relationships meant that loneliness resulting from relationship breakdown or the fear of having a relationship were also reflected in discussions. Comments such as a woman stating she felt 'really lonely and isolated' and another who said "you live a lonely life. People don't understand, so you live a lonely life". This lack of understanding was reflected by a further woman saying that "I was emotional and I was always throwing up and things and always complaining about pain, well it ends up pushing people away from me."

The relationship with intimate partners was the most discussed. There was a sense of men not being able to understand or empathise.

Men don't understand what we're going through because men don't go through it. (TI2)

Women explained how their HMB had affected their desire to have sex, either by causing them embarrassment or pain. Of course, there may have been other aspects in the relationships that were not working but even if that was the case, women's perceptions were that the

relationships did not work because of the HMB. This was a further knock to self-esteem and self-worth.

> I've lost two marriages. The last one was bold enough to go and ask a friend pharmacist about me having pain and not being able to have sex. And he didn't confront me. (WP5)

> So, I never married. I had boyfriends but I never married because I am too embarrassed for that (sex). (WP4)

An intimate couple attended our workshop. They were initially in separate small groups, so we were able to explore how their relationship had been affected by HMB, from the view of both partners. Both partners admitted to their relationship being affected.

> It does put pressure on our relationship obviously because she doesn't feel able to do as much she wants to do when she's in pain. (WP6)

> To be honest, you go to bed, you don't feel that sexy with loads of pads and your knickers on. (WP1)

For this couple, his presence at the workshop and his attitude towards her and other women with HMB, implied that their relationship was cohesive and supportive. Despite that, there was still an element of feeling 'pushed away'

> I feel well equipped to support her emotionally. The hard part is will she let me, she doesn't always let me. (WP6)

Thus, the theme of relationship breakdown and isolation were reflected even in a relationship that seemed very strong. Relationships with family were also affected, particularly the feeling of understanding and not only in male relatives, but also the difficulty of maintaining physical contact.

> I think my female relatives were actually less understanding than my partner was she'd (my mum) just be like oh, just get over it. Or my nan would be like oh, what are you moaning for, it's just a period. (WP1)

Another woman reported that her family members were supportive to the extent that they knew that she wanted to be alone and that her mood was affected by her periods.

### Pain

Pain and HMB are two separate symptoms but for women, they were so closely intertwined that pain became an important theme of our work. Some underlying causes of HMB also cause dysmenorrhoea (period pain), for example, adenomyosis. However, just passing large blood clots causes pain as the uterus contracts to expel the clots. Women expressed how pain was often the factor that stopped them from being able to conduct their daily activities. It was also recognised that being unable to manage pain exacerbated the feelings of helplessness.

> The pain stops me from doing my daily activities… for the first three to four days (of my period) I can't go to work, I can't go to uni, I can't do none of that stuff. (TI3)

> I'm miserable because I'm in pain and no one else can help me. (TI2)

Women were not describing 'period pain' per se, they were describing excruciating pain to the point of needing to attend hospital in an emergency. They were comparing it to childbirth and describing how even prescription only painkillers were at times still not enough to help them. Again, there seemed to be a lack of understanding when seeking help with women feeling 'belittled'.

> I just feel like if this was pain anywhere else, like in my chest, I would've been treated…like a person with pain, rather than just an over emotional woman. (WP1)

This lack of compassion from medical professionals was a common thread within our next theme 'perceptions of care'.

### Perception of care

How women perceived aspects of their care was echoed in all groups and interviews. Positive and negative reports were heard. Key factors were communication with medical professionals and discrimination; individualised care, with patients wanting to feel involved in their treatment choices and how they might be tailored to them, and the side effects of treatments.

Many women report that they were dismissed by medical professionals either being told that something was 'just a period' or normal for their age. Research has shown that a positive doctor–patient relationship results in better treatment and satisfaction outcomes,[17 18] thus it is disappointing to find that this was not most of our participants' experience.

> I think it's quite dehumanising when you go to a doctor's office and the doctor looks at you and goes what's your problem? You've got ten minutes, go. And you're sitting there you go well, I feel silly for saying this now, I mean don't know because I don't have a medical degree. So, I feel like a lot of the time women are patronised or intimidated. (WP1)

> I've been to the doctor, and they've not really been supportive. Just like, "Oh, you bleed a lot, here's some iron tablets. (TI1)

Many women perceived an element of discrimination in the approach of their doctor. Two aspects that were highlighted were age and ethnicity. A belief that it is normal for periods to get heavier and more irregular as you get closer to the menopause leads to many women being fobbed off, as explained by these participants:

One of the things for me as I got older was if I went to the GP, they'd say oh, it's your age. This is just normal and you'd think I'm sure it's not really. (WP5)

The thing is I'm not waiting until I turn 50 to say okay, well, menopause… They're waiting on the menopause to stop the problem. I want my life back now. (WP4)

Conversely, we did have a participant who felt that her age was a key factor in her prompt referral and investigation but she was clearly in the minority as no other participant had shared her experience.

My experience was a bit different because my GPs were really concerned about it. So, I was treated as first priority. (WP2)

Black women are more likely to develop fibroids than their white counterparts but they are also more likely to be diagnosed earlier and have more severe symptoms.[19 20] Our participants with fibroids were all black and echoed these data when discussing their diagnosis. They implied that their ethnicity had negatively affected prioritisation of fibroids and treatment options and was a factor in wanting individualised care.

it's the same treatment for everybody. But I'm telling you that a certain race of people, they suffer with it more bad and the treatments they have, this doesn't work. They should try and find … something to eradicate the fibroids or something at a young stage, from teenager. (WP4)

Ethnic black people have been suffering with this for years and there has been no solution for us. (WP5)

Individualised care was important to all women. They wanted to be involved in their treatment choices but it was also important to them to have as much information as possible about side effects and their individual risks of developing them. HMB has several underlying causes but many of these can be treated with the same treatment options. For example, hormones will treat many of the causes as would a hysterectomy but these options are very different in terms of their risks and effectiveness. Many treatments are either contraceptive or lead to the end of childbearing. With this in mind, it is easy to imagine why women want to be part of the decision-making process and why women with the same problem may well favour alternative treatments.

I was offered a hysterectomy and I declined because I felt it was quite a radical thing to do. (WP5)

I'm not comfortable with the idea of not having a period. Something that totally stops the period. I'm not comfortable with that. (TI1)

I think maybe if you feel you've been listened to…the options and the risks have been explained and maybe if alternatives are offered. So, 'we could go this way, or we could go that way', rather than 'we're going to go this way'. (WP1)

Hand in hand with that decision-making process was the information given regarding side effects of treatments. A key element of side effects was 'what is the payoff?'; what will I have to put up with if I have this treatment and am I willing to do it? Acknowledging the payoffs demonstrated how different subgroups of women may accept different payoffs and highlighted the desire for individualised care.

But I remember thinking actually if I was younger, it sounds weird, but I wouldn't want my periods to stop. But actually, because I was older, I thought that's fine. (WP2)

So, I think the side effects are sometimes a lot worse [than the bleeding] …. other women couldn't care less about the hair loss, whereas younger groups may care more about hair loss or acne as side effects. So I feel like that should be something that should be dedicated to, because I don't think one treatment can solve all problems. (WP1)

I would definitely want to know the advantages and disadvantages for example, when I was in the pills before, I was always worried about what if I'm on the pill for too long and then later on I can't conceive. (WP3)

These findings echo those of *a* systematic review of interventions for shared decision-making in HMB.[21] The authors identified that a third of patients felt that they were not given enough information about the advantages and disadvantages of treatments and that 74% of women wanted to make the decision about treatment with their doctor (as opposed to the 4% who wanted the doctor to choose and the 2% who wanted to choose on their own).

## DISCUSSION

The overwhelming theme that came out of this workshop was that of 'restriction'. All the other themes link back to how HMB stops women living their lives as they would wish. It affects their relationships, education, careers, reproductive wishes, social life and mental health. This is a condition of girls and women in the prime of their lives, at ages when they should be free to thrive and do whatever they want but for many, the constant threat of a heavy period starting and shaming them, means that they sacrifice that freedom. The societal and economic costs of women being incapacitated for a week or longer every month effects them, their friends and family and society as a whole.

A 2007 systematic review estimated that women with abnormal uterine bleeding (AUB) had on average a Health-Related Quality-of-Life Score below the 25th percentile of that for the general female population within a similar age range.[22] (AUB includes women with bleeding between their periods and after the menopause but the studies used in this analysis mainly included those with HMB.) They also estimated that the annual direct and indirect economic cost of AUB in the USA

is approximately US$13 billion.[22] With that in mind, the global economic burden of heavy bleeding must be exorbitant.

The aim of this work was to look for treatment outcomes that were priorities for patients but that had not been identified by our systematic review of previously used research outcomes. We identified five additional outcomes, all of which reflect the themes identified in our workshop. Cost is often considered during research but not in the context of the monetary costs that patients are subjected to. It is not just the cost of sanitary protection that needs to be factored in but clothing, bedding, towels, mattresses, pain killers, underwear and even loss of earnings. It could be argued that 'shame and embarrassment' could have come under 'psychological well-being', but they were so prominent within the discussions, we felt that they should be separated out as outcomes in their own right. Further, separating this out was better able to highlight the interplay between self-identity and social identity.[23] Bleeds are highly visible and occur within a group that is often marginalised anyway (often for religious or cultural reasons), moreover bodily fluids and staining, both further affect social identity—as someone who is contaminated and who damages property. The women's social identities led to reduced well-being as they were constantly anxious and in a state of hypervigilance.

We were able to compare findings from Dutch and Chilean women who had participated in similar workshops in their own countries. These workshops were independently run by collaborators, local clinicians, based on our design. Their contribution allows us to evaluate how opinions may differ in these countries. In the Chilean workshop, women's priorities were in keeping with those identified in our UK work. The most important aspects for them were trying to hide their pads and wearing dark clothes, as well as having to always be prepared by carrying pads and spare clothing and the potential for shame or embarrassment should they leak onto their clothes or onto furniture (eg, bus seats). They also mentioned restrictions on their social lives and sex lives. Public toilets are very hard to find in Chile, resulting in women with HMB avoiding public places during their periods. Cost was an important issue for them as they felt they were spending an enormous amount on pads and tampons. Like our population, what they wanted was a treatment that would remove restrictions and improve their ability to have a normal life at work and home. Side effects were not identified as particularly important to this group.

The Dutch workshop revealed the same findings regarding limitations on work and on physical, social and leisure activities. As a contrast to both the UK and Chilean workshops, patients did not regard small transient side effects or complications to be important, but their partners did. However, in keeping with UK women, the patients were concerned about external characteristics such as weight gain, acne and hirsutism. When considering menstrual blood loss, they were more interested in whether treatment returned their bleeding to normal levels than whether it stopped their bleeding completely (amenorrhoea—a common term used in research studies of HMB) and predictability was also prioritised. As with the UK workshop, individualised care was highlighted, with them wanting to know the experiences of others who are in the same 'situation' and have similar characteristics for example, age.

Although we only had a small number of participants in our workshop and our breakout groups were smaller than we had desired, we were able to supplement our work with interviews and identify that we had reached data saturation. In addition, data from international workshops was generally in keeping with our data which confirms the validity of our results. All the common underlying causes of HMB were represented by women in our workshop and they reflected a range of ages and ethnicities. Our UK work was limited by the participation of just one male partner of a woman with HMB. We identified that having a female researcher interviewing this participant may have influenced his answers because when his partner was discussing her problems in a separate group, she was more open about aspects of their relationship. We may have gained a more in-depth view, had we had a male researcher work with the male participant. We would have liked more input from other partners and representation from same-sex partners to identify any additional views. It may be that this reflects the opinions expressed about men 'not understanding' or the common view that it is 'women's business'. The Dutch workshop was attended by several partners, possibly representing the more liberal attitude to sex in the Netherlands.

## CONCLUSION

This work highlights that women with HMB face many restrictions which affect all aspects of their lives. Treatments, as well as symptoms, caused them distress and there was a general feeling of lack of empathy from close associates and health professionals.

We identified aspects of HMB which have not been previously reported as research outcomes and evaluated these to create five new outcomes to include in the next stage of core outcome development.

By conducting research with patients, we have allowed them to openly discuss HMB and give them a voice to explain how they are affected. Their input into our work is invaluable and will ensure that future research asks questions that are applicable to women.

**Acknowledgements** We would like to acknowledge the contribution of Dr Johana Nayoan, at the time a research fellow for Blue Communities Research Programme, University of Exeter, who facilitated the patient workshop. We would also like to thank Rosa Papadantonaki and Magdalena Bofill for sharing their international workshop results with us. Also, we thank Katie's Team (www.barc-research.org/katies-team) who advised us during development of our patient documents.

**Contributors** NAMC, KSK and CR developed the methodology, secured funding and ethical approval. NAMC, SY and AT recruited workshop participants. NAMC and CR facilitated the patient workshop, supported by AT and SY. NAMC and SY performed

data analysis. NAMC wrote the manuscript and acts as guarantor. All authors were involved in reviewing and redrafting the manuscript.

**Funding** The core outcomes set for HMB is funded by an Academy of Medical Sciences Starter Grant for Clinical Lecturers which was awarded to NAMC. NAMC is an academic clinical lecturer funded by the National Institute for Health Research. KSK is a distinguished investigator funded by the Beatriz Galindo (senior modality) Programme Grant given to the University of Granada by the Spanish Ministry of Education.

**Competing interests** None declared.

**Patient and public involvement** Patients and/or the public were involved in the design, or conduct, or reporting, or dissemination plans of this research. Refer to the Methodology section for further details.

**Patient consent for publication** Not applicable.

**Ethics approval** This study involves human participants. Ethical approval was granted by East Midlands Nottingham 1 Research Ethics Committee on 14 December 2015, REC reference ID 15/EM/0565. Participants gave informed consent to participate in the study before taking part.

**Provenance and peer review** Not commissioned; externally peer reviewed.

**Data availability statement** Data are available upon reasonable request. The data that support the findings of this study are available from the corresponding author upon reasonable request

**ORCID iDs**
Natalie Ann MacKinnon Cooper http://orcid.org/0000-0003-1929-833X
Khalid Saeed Khan http://orcid.org/0000-0001-5084-7312
Carol Rivas http://orcid.org/0000-0002-0316-8090

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
