## [Reviewer comments · BMJ Open]

ARTICLE DETAILS

TITLE (PROVISIONAL)	A QUALITATIVE STUDY EXPLORING WHICH RESEARCH OUTCOMES BEST REFLECT WOMEN'S EXPERIENCES OF HEAVY MENSTRUAL BLEEDING: STAKEHOLDER INVOLVEMENT IN DEVELOPMENT OF A CORE OUTCOME SET
AUTHORS	Cooper, Natalie; Yorke, Sarah; Tan, Alex; Khan, Khalid; Rivas, Carol

VERSION 1 – REVIEW

REVIEWER	Bannow, Bethany Samuelson Oregon Health & Science University
REVIEW RETURNED	07-Jul-2022

GENERAL COMMENTS	Cooper et al report on findings in their qualitative study of patients and partners of patients with HMB. Findings include a significant impact of HMB on quality of life, relationships, education, careers, reproductive wishes, social life and mental health. This is a hugely important and well done investigation and this reviewer applauds the authors on studying this topic and reporting their findings. I have two comments for consideration. The authors state that an inclusion criterion was having diagnosed HMB. Is there any standard diagnostic tool the referring providers used or was it subjective? Did the authors consistently collect data on heaviness from participants (at least one participant reported changing protection every 25-30 minutes, was that a standard question or simply volunteered by the participant). If this data is available it might help add perspective. It would be ideal if the authors could provide more specific guidance on how to measure these outcomes in future research. The additional guidance on costs to measure was helpful but might the authors provide some practical suggestions on, for example, how to “measure” or “report” restrictions or impact on relationships, among others? Knowing which outcomes are important to patients is huge but practical suggestions will eventually be needed in order to incorporate this valuable data.
--

REVIEWER	Connor, Mary Royal Hallamshire Hospital, Gynecology and Obstetrics
REVIEW RETURNED	16-Nov-2022

GENERAL COMMENTS	The study should provide useful information that can be used in future research into HMB. The focus on patients' concerns
---

	addresses a gap in knowledge and understanding as to what is important about the HMB condition and its management to patients. The source of funding for the study is not clear. It is not referred to if the paragraph titled 'Ethics and funding'. Other information, such as ethical approval and trial registration were adequately described.
--	--

REVIEWER	Chew, Kah Teik Pusat Perubatan Universiti Kebangsaan Malaysia, Obstetrics & Gynaecology
REVIEW RETURNED	09-Jan-2023

GENERAL COMMENTS	Very interesting work by all authors. Few issues required further clarification:  1. A total of 16 recruited participants who supposed to attend the workshop, only 6 turned out. What happen to those who did not attend the workshop? 2. The remaining 25 recruited participants who could not make it for the workshop, why only three were interviewed via telephone?
--

VERSION 1 – AUTHOR RESPONSE

Reviewer 1	
The authors state that an inclusion criterion was having diagnosed HMB. Is there any standard diagnostic tool the referring providers used or was it subjective? Did the authors consistently collect data on heaviness from participants (at least one participant reported changing protection every 25-30 minutes, was that a standard question or simply volunteered by the participant). If this data is available it might help add perspective.	The definition of HMB is subjective as per the UK NICE guideline, FIGO, and the American College of Obstetrics and Gynecology (ACOG) (1-4). Thus having been diagnosed in the NHS clinic was enough to be included and was based on the subjective definition in the NICE guideline.  1. National Institute for Health and Care Excellence. Heavy menstrual bleeding: assessment and management. 2018. Available from: https://www.nice.org.uk/guidance/cg44. Accessed 02/09/2022. 2. Munro MG, Critchley HOD, Fraser IS, FIGO menstrual disorders committee. The two FIGO systems for normal and abnormal uterine bleeding symptoms and classification of causes of abnormal uterine bleeding in the reproductive years: 2018 revisions. International Journal of Gynecology & Obstetrics. 2018; 143:393-408. 3. ACOG reVITALize Gynecology Data Definitions website. Available from: https://www.acog.org/practice-management/health-it-and-clinical-informatics/revitalize-gynecology-data-definitions. Accessed 02/09/2022. 4. Sharp HT, Johnson JV, Lemieux LA, SM C. Executive Summary of the reVITALize Initiative: Standardizing Gynecologic Data Definitions. Obstetrics & Gynecology. 2017;129(4):603-7.
It would be ideal if the authors could provide more specific guidance on how to measure	This paper describes the qualitative work from a project to develop a core outcome set for HMB. This defines outcomes that should be reported in all future studies of HMB. The paper

these outcomes in future research. The additional guidance on costs to measure was helpful but might the authors provide some practical suggestions on, for example, how to “measure” or “report” restrictions or impact on relationships, among others? Knowing which outcomes are important to patients is huge but practical suggestions will eventually be needed in order to incorporate this valuable data.	reporting the results of this work is currently under review by another journal.
Reviewer 2	
The source of funding for the study is not clear. It is not referred to if the paragraph titled 'Ethics and funding'. Other information, such as ethical approval and trial registration were adequately described	I am sorry. This was an oversight. The funding is described on page 18, however I have now added the following statement under the heading 'Ethics and Funding' The project was funded by an Academy of Medical Sciences Starter Grant for Clinical Lecturers, awarded to Dr Cooper.
Reviewer 3	
1. A total of 16 recruited participants who supposed to attend the workshop, only 6 turned out. What happen to those who did not attend the workshop? 2. The remaining 25 recruited participants who could not make it for the workshop, why only three were interviewed via telephone?	We emailed those who had not been able to attend to ask if they would agree to telephone interviews. Only three participants responded but all of them agreed to interview. The methodology section has been amended on page 4 to explain the email approach for interview: 'We emailed the recruited participants who had not attended the workshop, asking if they would be willing to undergo a telephone interview. Those who agreed were interviewed by the clinical academic researcher (NAMC)...' The results section has also been edited on page 6 'Three participants who could not attend the workshop agreed to undergo telephone interviews.....'

Editor	
We are concerned that the data presented in the paper has not been sufficiently anonymized. Table 1 presents exact ages of participants which is considered as a direct identifier. Unless you have obtained written informed consent from all participants to publish this information can you please ensure that the paper contains no direct identifiers and no more than two indirect identifiers? We suggest presenting the data in table 1 in aggregated form or using age ranges.	Thank you. Table 1 as been edited with age ranges instead of exact ages.